# Pattern of Urban Flora in Intra-City Railway Habitats (Alexandria, Egypt): A Conservation Perspective

**DOI:** 10.3390/biology10080698

**Published:** 2021-07-22

**Authors:** Selim Z. Heneidy, Marwa W. A. Halmy, Soliman M. Toto, Sania K. Hamouda, Amal M. Fakhry, Laila M. Bidak, Ebrahem M. Eid, Yassin M. Al-Sodany

**Affiliations:** 1Department of Botany & Microbiology, Faculty of Science, Alexandria University, Alexandria 21511, Egypt; drsheneidy@yahoo.com (S.Z.H.); smt_toto@yahoo.com (S.M.T.); drsaniakamal@gmail.com (S.K.H.); amalfakhry@live.com (A.M.F.); laila_bidak@yahoo.com (L.M.B.); 2Department of Environmental Sciences, Faculty of Science, Alexandria University, Alexandria 21511, Egypt; marwa.w.halmy@alexu.edu.eg; 3Biology Department, College of Science, King Khalid University, Abha 61321, Saudi Arabia; ebrahem.eid@sci.kfs.edu.eg; 4Department of Botany & Microbiology, Faculty of Science, Kafrelsheikh University, Kafr El-Sheikh 33511, Egypt

**Keywords:** railway flora, human disturbance, alien species, ruderal species, plant communities, endemic species, urban habitats, Alexandria

## Abstract

**Simple Summary:**

Intra-city railway areas can play a role in enhancing the diversity and dynamics of urban flora. The current study sheds the light on the role that overlooked railway habitats can provide in the conservation of biodiversity through surveying the floristic composition and diversity along intra-city railway and tram tracks in Alexandria. The plant communities were identified using multivariate analysis techniques. Spontaneous flora in intra-city railway areas represent distinct adaptations to unique urban–industrial ecosystems with different levels of anthropogenic disturbance. Despite the high level of disturbance, native species dominated the investigated habitats, including rare and endemic species. The study emphasizes the role that these ruderal habitats provide as valuable refuge areas for rare and endangered species worthy of proper management and conservation action.

**Abstract:**

Intra-city railway areas are deemed large greenspaces and are believed to be key in enhancing the diversity and dynamics of urban flora. In the current study, the floristic composition and diversity along intra-city railway and tram tracks in Alexandria were surveyed. The floristic composition of the plant communities in relation to environmental factors, ecological indicators, and level of human impact was analyzed using multivariate analysis (two-way indicator species analysis (TWINSPAN) for classification and detrended correspondence analysis (DECORANA) for ordination. The multivariate ordination techniques (CCA) revealed differences in the environmental factors and climatic factors influencing the floristic composition of the railway and tram track habitats. Tram tracks suffered higher human impact while maintaining higher vitality and cover compared to train tracks. Species recorded were mainly therophytes, followed by phanerophytes and hemicryptophytes dominated by native species; however, invasive species’ contribution was higher compared to surrounding regions. The number of invasive species was greater in railway areas compared to tram track areas (19 and 15, respectively). The occurrence of two endemic species (*Sinapis allionii* and *Sonchus macrocarpus*) with limited national distribution highlights the importance of these habitats as valuable refuge areas for rare and endangered species worthy of conservation action.

## 1. Introduction

Human activities over the last few centuries have resulted in immense modifications in natural landscapes and the formation of new human-made habitat types [1]. Many studies have examined the floristic composition of urban areas in a variety of regions and at different scales [2,3,4]. These studies have provided basic knowledge of the main factors influencing species richness and floristic composition in urban habitats [5,6]. Urban ecology has a long history, and the field has been rapidly developing in the last few decades [7,8], through increasing surveys describing the vegetation composition of urban habitats [9,10,11,12]. Investigations in urban habitats have found these landscapes to be rich in species owing to the diversity of habitats that they include [13], in addition to their enrichment by invasions of alien species [6]. The human-made habitats include irrigation and drainage canals, railways, motor roads, railway yards, demolished houses, abandoned fields, refuse areas, and graveyards. The construction and use of tracks, roads, canals, railways, and airports have caused many direct and indirect changes in these urban centers. Direct influences include the destruction of the existing habitats and the provision of new ones with special characteristics. The tracks, roads, canals, and railways provide continuous stretches of open habitats extending for hundreds of miles and forming a nation-wide network, offering more opportunities for species’ rapid colonization and spread. Some alien species take advantage of these networks; however, more attempts are needed to study their effect on native plant communities [14].

In a rapidly urbanizing world, understanding the response of ecosystems to urbanization is needed to ensure that urban areas are included in planning for the sustainability of residents and nature [15]. Urban areas and the human population are growing progressively; with the growth of urbanization, human disturbance is increasingly affecting biodiversity including urban flora [16]. Anthropological activities may create various specific habitats suitable for the recruitment of some plant species. The study of the urban habitats in Egypt has attracted considerable attention in the last few decades. For example, the early study of Shaltout and Sharaf El-Din [17] along the Cairo–Alexandria agricultural road led to the identification of seven habitat types and 19 plant communities. Shaltout and El-Sheikh [18] studied the species diversity of the urban habitats in the Nile Delta and recorded 248 species representing 46 families.

Despite the numerous studies that have addressed the ruderal vegetation and alien species of the Egyptian habitats, no study has accounted for the flora of Egypt’s major urban centers, such as great Cairo and Alexandria. Alexandria, one of the oldest cities existing, is named after Alexander the Great, who founded it in 331 BC [19]. Since its foundation in the Roman era more than three thousand years ago, the city has served as a melting pot for many Mediterranean as well as global cultures [20]. Alexandria is considered the most multi-cultural and exceptionally diverse city in Egypt. The integration of these cultures has made Alexandria an approachable and livable city for different nationalities and ethnicities [21]. Alexandria has always exhibited an elegant urban environment due to the culturally rich and diverse eras that the city has encountered [20]. Understanding how urbanization influences vegetation at Alexandria is essential because these vegetation communities provide various critical ecosystem services and maintain a high level of biodiversity [22]. The city is facing high risks due to natural and human-induced threats [23]. On one hand, the environment of the city has suffered heavily due to the increasing population, rapid urbanization, industrial expansion, and the increased utilization of fossil fuels in automobiles, trucks, and public transportation, resulting in high levels of pollution, worsened by the warm climate [24]. On the other hand, recently, the city has been experiencing extreme weather events including flash floods and storm surges, in addition to other consequences thought to be related to global warming, such as shoreline erosion, saltwater intrusion, and the threat of accelerated sea level rise [23].

The sides of roads and railways provide open spaces in urban habitats that facilitate the colonization and invasion of plant species to surrounding natural habitats including the deserts. Records exist also indicating a reverse trend, where some desert plants have managed to invade urban habitats [25]. These high-stressed microsites are places with low competition, which allow these plants to become established [26]. Urban habitats, particularly roads and railways, provide habitats for mixed flora known as anthropogenic–azonal vegetation [27]. Knowledge and understanding of the site history facilitate the evaluation of the influence exerted by humans on shaping local flora [9]. Tram and train tracks are synanthropic habitats that host specific types of flora. Plants growing on both tram and train tracks must adapt to this specific environment, resulting in the development of unique characteristics [28]. Tram transport represents one of the major types of public transport in Alexandria, and the tracks pass through almost all parts of the city. Railways generate noise and vibrations, producing large amounts of emissions and toxic substances that cause pollution (air, soil, and water), which may influence species’ abundance and richness [29,30,31]. Therefore, plants inhabiting such habitats are usually stressed by traffic emissions, polyaromatics, persistent organic pollutants, etc. Plant species growing in these habitats can endure severe mechanical disturbances or toxic materials.

The construction and use of tracks, roads, canals, railways, and airports have involved many changes—some of them are direct and others are indirect. Direct influences include the destruction of the existing habitats and the provision of new ones that have special characteristics. These have provided more or less continuous stretches of open habitats extending for hundreds of miles and forming a nation-wide network, with opportunities for rapid colonization and spread. A few alien species have taken advantage of these networks, but no serious attempts have been made to study their effect on native plant communities [14]. The well-drained stony habitats of railway and tram tracks offer appropriate requirements for the growth of xero-thermophilus synanthropic species [4], which represent an abundant component of the floristic spectrum of the railway and tram track habitats investigated in the current study. However, numerous species of the adjoining habitats usually infiltrate onto tracks [32,33] and can be relatively abundant, including seedlings of woody shrubs, which contributed to the floristic spectrum of railway habitats—for example *Arthrocnemum macrostachyum* (Moric.) C. Koch, *Atriplex halimus* L., *Atriplex leucoclada* Boiss., *Sarcocornia fruticosa* (L.) A.J. Scott and *Suaeda pruinosa* Lange. Similar studies revealed similar cases [28,34,35,36,37]. Seedlings and juveniles of woody plants and other different life forms contribute to the floristic spectrum of rail transportation areas, which led to some deviation for different vegetation criteria. Most of the research was dedicated to the flora and vegetation of railways [38]. Studies related to the vegetation and flora of railways have been carried out in many places all over the world [2,4,33,35,37,38,39,40]. In local environments, these habitats can also serve as valuable refuges for some endangered and rare species [33]. The floristic spectrum of the green train or tram tracks is often controlled by land-use planning, and it must be involved in the management of the surrounding area [41].

Although tram and railway tracks represent specific urban habitats and host a specific type of flora, the vegetation of tram and train tracks in cities is poorly studied, particularly in the arid region’s cities. Knowledge of the floristic spectrum growing on tram tracks is just as important in terms of the potential future urban planning and land-use management of cities. Tram transportation represents one of the major means of public transport in Alexandria, and the tracks run through almost all parts of the city, connecting its extremities from west to east. The current study provides a documentation of the urban flora and the floristic composition of Alexandria’s intra-city railways and tram tracks. The study will also attempt to assess how human activities, through the fragmentation of natural habitats and the construction of buildings and roads, have shaped the city’s plant diversity. Specifically, the study aims to: (1) examine the pattern of the floristic composition in the urban habitats, especially tram and train tracks, of Alexandria; (2) evaluate the effects of human impact activities along these habitats on the plant diversity; (3) categorize the alien and native taxa and the main plant communities prevailing in the city and detect the key factors influencing their distribution; and finally (4) identify elements of the city’s flora of conservation significance and provide information on their spatial distribution to help in taking appropriate measures to sustain their existence and conservation.

## 2. Materials and Methods

### 2.1. Study Area

The current study focuses on Alexandria, located to the west of the Rosetta branch of the Nile River (Figure 1). It is the largest city on the Mediterranean, also known as the ‘Bride of the Mediterranean’ and the second largest city in Egypt. In ancient times, during the Roman era, the city was a flourishing commercial center with a population of a few hundred thousand [42,43,44]. Currently, the city is the main harbor in Egypt, and by hosting nearly 37% of Egypt’s industries (large oil refineries, chemical, cement, metal plants, textile mills, food processing operations), it is deemed the second most important industrial center in Egypt [45]. The city has a population of more than five million [46] and is considered the most popular summer resort for Egyptians. Alexandria Governorate is distinctly characterized by unique variations in its geomorphology, topography, land cover, and shoreline stability. The coastal plain is characterized by the existence of a lowland relic depression surrounded partly by high-elevated limestone ridges aligned parallel to the coastline [23].

The location of the city and the predominant north wind gusting throughout the Mediterranean provide Alexandria with a distinctly unique climate compared to that of the surrounding areas and the desert hinterland. The summer is comparatively temperate, even though humidity can markedly increase in July and in August, the warmest months, while winter is cool and regularly marred by a sequence of strong storms that are usually accompanied by heavy rain and sometimes hail. Air temperature fluctuates seasonally, recording a maximum average of 36.6 °C during summer and a minimum average of 8.2 °C during winter. The average maximum temperature is 34.7 °C and the average minimum temperature is 14.7 °C. The mean annual wind speed anomaly ranges from −3.03 kt to 1.28 kt. It has an increasing trend with a rate of 0.125 kt/year. The mean annual amount of precipitation is 178.63 mm with a precipitation rate of 5 mm/rainy day [47].

### 2.2. Sampling and Data Collection

Fifty-one stands were selected along transportation railways (24 stands along train and 27 stands along tram tracks) during the spring season 2020 (Figure 1, Table 1). The stand size varied according to the habitat type and the extension of plant cover (which approximate the minimal area of the plant communities). In each stand, the following data were recorded: (a) location in latitude/longitude coordinates using handheld Garmin GPSMAP^®^ 64s, (b) list of natural, cultivated trees and weed species, (c) the most dominant species, (d) a visual estimate of the total cover (%) and the cover of each species according to the Braun-Blanquet scale [48], and (e) the human activities and disturbance occurring in each stand (e.g., pollution, firing, grazing, overcutting, etc.) The data were compiled into a raw table containing the recorded species in the examined stands. Duplicate specimens of the present vascular plants were collected during each visit. The herbarium specimens prepared and identified nomenclature was according to Boulos [49,50] and is presented in the author’s collections.

### 2.3. Vegetation Measurements

Life forms of the species were identified following the Raunkiaer scheme [51] as follows: Phanerophytes, Chamaephytes, Geo-Helophytes, Hemicryptophytes, Parasites, and Therophytes. By determining the contribution of the various life forms to the flora of a certain region, one can obtain an ecological spectrum or lifeform spectrum which reflects a certain relationship with the climate of an area. The chorotype of the recorded species was determined from Zohary [52] and Feinbrun-Dothan [53], and the local distribution according to Täckholm [54] and Boulos [49,50]. The threatened species were determined according to the IUCN (1998) Red list categories as reported by El-Hadidi and Hosni [55].

### 2.4. Environmental Variables and Measures of Disturbance

To examine the level of anthropogenic impact in the study area and how this influences the species composition, a set of variables were assessed at the stand as indicators of the degree of anthropogenic influence. The decision to include the parameters of disturbance prevailing in the region was based on observations during the field studies. These variables included the road density (m/m^2^) and the distance from the coast. Geodata from the OpenStreetMap database [56] were acquired for Alexandria and were used for the estimation of the distance to the coast and the road density. The road density was calculated as the average length of roads in meters per m^2^ in 500 m buffer zone surrounding each stand. Analyses of the geospatial data and derivation of the geospatial indicators were conducted within the framework of ArcGIS package 10.1 [57]. Climate data were extracted for the sampled site from the WorldClim 2 data [58].

### 2.5. Assessment of Human Impact

The degree of artificialization expresses the magnitude of human, domestic animals, and urbanization levels on the ecosystems. In general, the pressure is very weak in areas far from urban centers and where the use of resources is such that man does not influence the environment to a notable degree [16]. Using the recorded field data and observations on human activities and threats, the level of human interference (L) was classified into four levels as follows: 0 = no interference, 1 = low interference, 2 = medium interference, 3 = high interference. These levels were assigned according to the scale for the degree of human interference adopted by Ayyad and Le Floch [59]. The impact level score of each threat was assigned in each stand considering the state of vegetation cover, the vitality of the species, which refers to their capacity to live or grow, species richness, and the number of alien species, weed, and native plant species.

For each stand, the intensity of human impact *HI* was calculated using the following equation: $HI=\sum_{i}^{n}L*i$. Here, *i* is the type of threat (see Appendix A for a list of all threats detected in the sampled sites), *n* is the number of different threats recorded at each site, and *L* is the level of interference caused by threat *i* at each site.

### 2.6. Diversity Indices

Diversity is function of species richness and the equitability of individual distribution (i.e., evenness) amongst the species [60]. Five of the more popular indices of alpha diversity were applied according to Magurran [61] as follows: (1) Richness = number of species recorded in each site; (2) Shannon’s diversity index (H′) = − ∑ p_i_ ln p_i_, where *p_i_* is the proportional abundance of the *i_th_* species; (3) Simpson’s index of dominance (D) = ∑ p_i_^2^; (4) Hill’s number 1(N1) = exponential Shannon’s index = e^H′^, and (5) Shannon evenness index (E1) = H′/H_max_ = H′/ln S, where *S* is the number of species.

### 2.7. Data Analysis

Both the two-way indicator species analysis (TWINSPAN) and detrended correspondence analysis (DECORANA) were carried out to the cover estimations of the 224 species recorded in the sampled 51 stands to identify the plant communities in the study area [62,63,64]. Simple linear correlation coefficients between diversity indices and environmental factors were determined. Before analysis, the data were evaluated for normality of distribution and homogeneity of variance using Shapiro–Wilk’s W test and Levene’s test, respectively. Log transformation was performed on the data when necessary, prior to conducting analysis of variance (ANOVA). One-way ANOVA (ANOVA-1) was used to identify statistically significant differences in components of diversity indices between the 6 vegetation groups, while Student’s *t*-test was used to identify significant differences among two habitats (train/tram). All statistical analyses were performed using SPSS software [65]. For analyzing the relation between environmental variables and ecological groups from direct gradient changes, the canonical correspondence analysis or CCA ordination was applied [66]. The CCA ordination examined the relation between species and environmental factors as a linear compound [67].

## 3. Results

### 3.1. Floristic Composition

The recorded species in the study area, their national geographical distribution, life forms, habitats, chorotypes, vegetation groups, and their relative constancy are listed in Appendix A. The total number of recorded species was 224, belonging to 155 genera and 51 families. Around 45.1% of the recorded species (101 species) were perennials, while 54.9% (123 species) were therophytes. One hundred and eight species (32 perennials and 76 therophytes) were recorded as weeds associated with the natural flora in the sampled stands, while 60 species (26.8%) were assessed as alien ones (30 casual, 25 naturalized, and 5 invasive species). These invasive species are: *Acacia saligna* (Labill.) H.L. Wendl., *Bassia indica* (Wight) A.J. Scott, *Dalbergia sissoo* Roxb. ex DC., *Ipomoea carnea* Jacq., and *Prosopis juliflora* (Sw.) DC. On the other hand, 12 species were common and recorded in ≥50% of the sampled stands: *Chenopodium murale* L. (84.3%), *Cynodon dactylon* (L.) Pers. (78.4%), *Malva parviflora* L. (76.5%), *Urospermum picroides* (L.) F. W. Schmidt (70.6%), *Sisymbrium irio* L. (64.7%), *Lactuca serriola* L. (58.8%), *Phoenix dactylifera* L. (58.8%), *Senecio desfontanei* Druce (58.8%), *Sonchus oleraceus* L. (54.9%), *Avena fatua* L. (52.9%), and *Hordeum leporinum* Link (51.0%). The first five families contributed 54.0% (121 species) of the total number of species in the following sequence: Poaceae > Asteraceae > Brassicaceae > Chenopodiaceae > Fabaceae (Figure 2).

Regarding the lifeform spectra of the recorded species, therophytes had the highest contribution (54.9% of the total species), followed by phanerophytes (15.6%), hemicryptophytes (15.2%), chamaephytes (9.4%), and geophytes (3.1%) (Figure 3).

The local geographical distribution indicated that 73.7% of the recorded species belonged to the Mediterranean coast, followed by the Nile region (64.7%), Egyptian deserts (64.3%), Sinai (53.6%), and Oasis (46.0%) (Appendix A). Regarding the global phytogeographical distribution, the mono-regional elements were the highest (95 species = 42.4%), followed by bi-regionals (63 species = 28.1%), while the pluri-regional (49 species = 21.9%) and cosmopolitans (15 species = 6.7%) were the lowest (Figure 4a). Thirty species (31.6%) of the mono-regionals were Saharo-Arabian, Mediterranean, and Tropical, which were represented by 18 species (19.0% each) and 14 species (14.7%), which were Sudano-Zambezian elements. In general, regarding either mono-, bi-, or multi-regional elements, 126 species belonged to Mediterranean, 78 to Irano-Turanian, 59 to Saharo-Arabian, and 42 to European elements (Figure 4b). It is worth noting that two endemic species were recorded in the studied urban habitat, namely *Sinapis allionii* and *Sonchus macrocarpus*.

### 3.2. Plant Communities and Diversity Indices

The application of TWINSPAN on the cover estimates of 224 species recorded in 51 stands led to the recognition of 18 groups at the sixth level of classification and six vegetation groups (communities) at the third level (Figure 5a). The application of DECORANA on the same set of data indicates a reasonable segregation among these groups along the ordination axes 1 and 2 (Figure 5b). The vegetation groups are named after the species that have the highest presence percentage as follows: I: *Urospermum picroides*, II: *Chenopodium murale*, III: *Malva parviflora*, IV: *Cynodon dactylon*, V: *Hordeum leporinum*, and VI: *Sonchus oleraceus* (Table 1). The first, second, third, and the sixth groups (I, II, III, and VI) represent the train railways, while the fourth and fifth ones (IV and V) represent the tram track habitats.

Regarding the variation in diversity components, the species richness, Shannon index, and cover differed significantly in relation to both vegetation groups and habitats. The group of *Urospermum picroide*s (I) had the highest Simpson index and evenness (0.04 ± 0.02 and 0.96 ± 0.01, respectively), but the lowest of Shannon index (2.77 ± 0.38), while that of *Hordeum leporinum* (V) had the lowest species richness, Shannon index, and visual cover (18.09 ± 2.98, 2.77 ± 0.18 and 16.55 ± 20.83%, respectively) and the highest Simpson and Hill index (0.04 ± 0.02 and 2.55 ± 3.05). The group of *Sonchus oleraceus* (VI) had the highest of species richness and Shannon index (38.67 ± 15.82 and 3.44 ± 0.39), but the lowest Simpson index, evenness, and Hill index (0.03 ± 0.01, 0.95 ± 0.00 and 1.37 ± 0.03, respectively). On the other hand, the train habitat had the highest species richness, Shannon index, evenness, and visual cover, while the tram habitat had the highest Simpson and Hill index (Table 2). It was indicated that group II (*Chenopodium murale*) had the highest total species (gamma diversity = 142 species), natural species, and number of native-weed and alien species (34, 95, and 24, respectively). On the other hand, group V (*Hordeum leporinum*) had the lowest gamma (60 species) and the lowest number of native-weed and alien species (32 and 15, respectively).

Regarding the environmental factors, the *Hordeum leporinum* (V) group represented sites characterized by high population density and rainfall (51816.9 ± 22813.4 ind. km^−2^ and 184.09 ± 0.30 mm, respectively), but the lowest temperature and distance from the sea (20.00 ± 0.00 °C and 630.76 ± 336.78 m); meanwhile, that of *Malva parviflora* (III) represented sites characterized by high altitude, population density, and rainfall (1.50 ± 1.91 m, 25715.5 ± 7216.3 ind. km^−2^ and 177.00 ± 3.37 mm, respectively). The group of *Sonchus oleraceus* (VI) was characterized by high temperature and distance from the sea, but the lowest road density (Table 3). On the other hand, the tram tracks were characterized by high altitude, population density, rainfall, and road density, while train tracks were characterized by high temperature and distance from the sea. The population density had a significantly negative correlation with species richness and species diversity as measured by Shannon’s index (*r* = –0.277 and –0.328, *p* < 0.05), but a significantly positive correlation with dominance as measured by Simpson’s index (*r* = 0.335, *p* < 0.05). The distance from the sea had a significantly positive correlation with species richness and Shannon’s index of diversity (*r* = 0.535 and –0.498, *p* < 0.001), but a significantly negative correlation with Simpson’s index of dominance (*r* = –0.354, *p* < 0.01) (Table 4).

Moreover, the correlation between the identified vegetation groups and the environmental factors is indicated on the ordination diagram produced by canonical correspondence analysis (CCA, Figure 6). The two groups that represented the train railways (I and VI groups) occupied the sites with high temperature, with medium to low population density and distance from the sea.

The other two groups that represented the train railways (II and VII groups) occupied sites with high population density and rainfall. On the other hand, the two groups that represented the tram tracks occupied sites with high rainfall and medium values of almost all investigated environmental factors.

### 3.3. The Impact of Human Activities

The results revealed that human impact activities had distinct effects on the flora associated with rail transportation. The application of the human impact formula (HI) on the collected data for both tram and train tracks indicated that the tram tracks exhibited higher levels of human impact (10 levels) than the train tracks (8 levels). Both plant cover and species vitality were decreasing with increasing human impact level at train and tram tracks. At tram tracks, it is obvious that at the highest level of recorded human impact (10), vitality recorded a high average value, exceeding the corresponding value for plant cover due to the presence of a high average number of alien species (4.8). The same trend was indicated at impact level 5, where the highest average of alien species (6.5) was recorded along the tram tracks. The other vegetation criteria—species richness and numbers of weeds, aliens, and natural species—fluctuated through the other human impact levels for both tram and train tracks. At tram and train tracks, species richness and the number of weeds exhibited the same trend along with the different human impact levels, while the numbers of alien and natural species indicated the same trend, except for impact level 5 at tram tracks (Figure 7).

## 4. Discussion

### 4.1. Floristic Composition

The species recorded in the present study (224 species) represent some 10.4% of the whole Egyptian flora, 20.5% of the genera, and 39.5% of the families represented in Egypt [50]. The area studied is a part of the northwestern coastal region of Egypt, which belongs to the Mediterranean phytogeographic region of the nation. This phytogeographic region is considered the richest in the nation for its floristic composition, owing to its relatively high rainfall. It contributes ca. 45% of the total flora, 57% of the genera, and 75.2% of the total families in Egypt [49]. Several factors influence plant diversity in the Mediterranean region, including topographic variability, seed banks, secondary succession, and temporal variations in climate.

The order of contribution of the five main families in the study area (Poaceae, Asteraceae, Brassicaceae, Chenopodiaceae, Fabaceae) is different to that recorded for the entire Egyptian flora (Poaceae, Fabaceae, Asteraceae, Brassicaceae, Caryophyllaceae), according to Boulos [50], and that reported for Sinai (Poaceae, Fabaceae, Asteraceae, Brassicaceae, Caryophyllaceae) as reported by Shaltout et al. [68]. Approximately 45.1% of the recorded species were perennials, while 54.9% were annuals. Regarding the lifeform spectra of the recorded species, therophytes had the highest contribution, followed by phanerophytes, hemicryptophytes, chamaephytes, and geophytes. The domination of annuals appears to be due to the warm–dry climate, topographic effects, and biotic impact [69]. The short life cycles of the annual weeds, in addition to the harsh climate and insufficient moisture, favored the presence of annuals during the favorable seasons. The occurrence of phanerophytes may be due to the escape of ornamental species from nearby gardens and hedges along the sides of both the train and tram tracks.

Out of the 136 alien species introduced to the Egyptian flora [70], 60 species were recorded in the present study (44.1% of the total alien species). The contemporary references of the Egyptian flora classified the alien species into five categories: escape, casual, introduced, naturalized, and invasive species [49,50,54,71,72]. It was estimated that the number of naturalized aliens is around 101 species [73]. The high numbers of naturalized alien species reported by Vilà et al. [73] in Egypt may be due to the evaluation of the species formerly listed as casuals. The 60 alien species recorded in the current study were categorized as 30 causal, 25 naturalized, and 5 invasive species, namely *Acacia saligna*, *Bassia indica*, *Dalbergia sissoo*, *Ipomoea carnea*, and *Prosopis juliflora*. *Acacia saligna* was reported as an invader in North Sinai by El-Bana [74]. *Bassia indica* was introduced into the northwestern coastal strip of Egypt from India in 1930 as a promising fodder plant and is now recognized as an aggressive invader [75]. *Ipomoea carnea* was introduced from South America for ornamental purposes in 1932 [76] and is currently recognized as a naturalized species along the water courses in the Nile Delta [77]. The aggressive invader *Prosopis juliflora* was introduced to the southeastern part of Egypt in 1952, for timber and charcoal production [75]. Moreover, *Dalbergia sissoo* was introduced as a source of timber and for ornamental purposes [78].

One hundred and eight species were recorded as weeds associated with the natural flora in the sampled sites, which explains the high contribution from family Chenopodiaceae to the urban flora of the intra-city railways. Egypt is an old nation with ancient civilization, which makes it difficult to determine with certainty if most of the currently available weeds belong to the native flora or were introduced intentionally or accidently. Similar findings were reported by Sheded and Shaltout [79] along the Red Sea coast of Egypt. There is no doubt that some weed species are native to Egypt and occur as elements of the natural vegetation. For example, *Anagallis arvensis* occurs in cultivated land and *Alhagi graecorum* is a ruderal species [70]. On the other hand, many of them were common segetal and ruderal weeds that were recorded to inhabit the Nile Delta [77]. Among these ruderals, *Chenopodium murale*, *Convolvulus arvensis*, *Cynodon dactylon*, *Cyperus rotundus*, *Eruca sativa*, *Malva parviflora*, *Polypogon monspeliensis*, *Portulaca oleracea*, *Sonchus asper* and *Sonchus oleraceus* were segetals and *Polygonum salicifolium*, *Polygonum equisetiforme*, *Cynanchum acutum*, *Phyla nodiflora*, *Datura innoxia*, *Plantago lagopus*, *Sylibum marianum*, and *Pluchea discorides* were thought to have been recruited here due to the land use change and the continuous modifications in the study area manifested by the fast establishment of residential buildings and green spaces in their surroundings. This, in turn, led to the arrival of a random weed species; meanwhile, the original native species have disappeared. The anthropogenic environments have different thermal ranges and moisture regimes compared to the surrounding natural regions; therefore, the selection of specific species with dispersal, physiological, and morphological characteristics is equipped to fill in the existing niches [80].

### 4.2. Plant Communities and Diversity Indices

Variations in the results of species diversity (Table 2) indicate high species diversity in all vegetation groups, with slight differences between different vegetation groups in the study area (Shannon’s index, H′ ranges between 2.77 to 3.44). This was accompanied by low dominance as confirmed by Simpson’s index (D ranges from 0.03 to 0.04). The number of equally common species (Hill’s number, N1) was estimated to be equal to around two to three species in the different vegetation groups. This clarifies the increase in the evenness of species’ relative abundance in the plant communities of the study area (Shannon evenness ranges between 0.88 and 0.96). It is worth mentioning that the *Hordeum leporinum* community (group V) was the least diverse, as confirmed by the applied diversity indices; Shannon’s index (2.77) and Shannon evenness index (0.88) coincided with the highest dominance as estimated by Simpson’s index (0.04), with around three equally common species sharing dominance. The highest number of species (142 species) was recorded in vegetation group II (*Chenopodium murale*), while the lowest numbers of species (58 and 60 species) were recorded in vegetation groups I (*Urospermum picroides*) and V (*Hordeum leporinum*), respectively. It is also notable that the *Chenopodium murale* community showed the highest numbers of both alien and weed species (37 and 70 species, respectively), while the lowest numbers of alien and weed species were recorded in the *Hordeum leporinum* community (15 and 32 species). Generally, each diversity index tested in the present study indicates higher species diversity of the train railways than the tram tracks, coinciding with the higher dominance in the tram habitat and higher plant cover in the train railways. Most species growing in these habitats can withstand the serious mechanical disturbances. As the tram tracks are frequently established near roads, the plants are stressed by traffic emissions, persistent organic pollutants, polyaromatics, etc. The soil near the tracks contains high levels of pollutants (e.g., oil residues) and heightened concentrations of heavy metals [38]; in many cases, the soil is dry and stony. Plant species that can grow in these habitats have to tolerate these unfavorable conditions. Sometimes, the vegetation expanding along the tracks is subjected to eradication using herbicides. It also has to deal with the trampling and mechanical disturbance of vehicle traffic [34,39]. Therefore, the weeds and invasive species always represent a considerable portion of species inhabiting these habitats. However, these habitats can still harbor rare and important species. Recording the two endemic species (*Sinapis allionii* and *Sonchus macrocarpus*) in the studied urban habitats confirms that railway and tram habitats may offer refuges for certain rare species. In addition, the vegetation in such habitats can serve several other different beneficial functions [40].

The considerable percentage of invasive and alien species (26.8%) recorded along railways in the current study reflects the role that these types of habitats play in spreading invasive species. Many studies show the great significance of the railway habitats in facilitating the spread of alien plant species [33,81], including invasive species [2,3,37], but the mechanism and the role of tram tracks in the alien invasions is not very well known so far. Goods and people transport also transfers plant diaspores, which may also be transported directly through being trapped on the vehicles. When constructing railway and tram tracks, new microhabitats are created in the vicinity. These microhabitats offer suitable sites for the spreading of alien species [81]. This might lead to the assumption that tram tracks could serve as a significant habitat for the spreading of alien species, despite occupying a smaller area as compared to the railways. The study of Denisow et al. [37] on railways has shown that both the richness and abundance of invasive alien species have been affected by the distance from the railway tracks and that this factor should be taken into account.

Despite the similarities in some characteristics of the habitats of railway and tram tracks, their flora may differ considerably [32]. In the current study, each habitat exhibited specific characteristics, which was reflected in the differentiation of the vegetation groups detected into two main communities that represented each of these habitats. The two groups that represent the train railways (I and VI groups) occupied the sites with high temperatures, with medium to low population density and distance from the sea. The other two groups that represent the train railways (II and VII groups) occupied sites with high population density and rainfall. On the other hand, the two groups that represent the tram railways occupied sites with high rainfall and medium values of almost all investigated environmental factors. In general, the tram tracks exhibited higher levels of human impact compared to the train tracks; on the other hand, the tram track sites attained high vitality and cover. The high plant cover may be due to the presence of a high average number of alien and weed species, which can thrive under disturbance conditions. Moreover, the dumping and accumulation of trash, debris, and refuse from nearby buildings, gardens, and factories can provide a potential source for nutrients in the urban habitats of the study area [82]. Moreover, a possible reason behind the high vegetation on the tram tracks is that the habitats of this region are usually affected by drainage water canals as well as from the adjacent roadside tracks. This process usually leads to the improvement of the moisture availability in these areas and favors the growth and vitality of species [83].

### 4.3. The Impact of Human Activities

Anthropological activities generate many specific habitats suitable for the growth of plants [84]. Although species richness shows a considerable decrease under the high disturbance levels in the study area, the general trend of plant species richness shows a remarkable increase under moderate disturbance, coinciding with the increase in the numbers of both alien and weed species richness. Both species richness and diversity are associated with increasing vegetation cover and positively correlated with an increasing distance from the seafront. The human impact in the study area may explain the high number of weeds in the recorded flora. This agrees with the assumptions of El-Fahar and Heneidy [85]. They also reported that the increase in alien plant species is linked to the human activities in the area. Population density negatively affects both vegetation cover and species diversity, while being positively correlated with the dominance of some plant species in different sites. The recorded field data showed that the same human activity varies in its effect on vegetation composition from one stand to another.

First, urbanization expands the area of impermeable soil, decreasing rainfall infiltration and contributing to the depletion of the original habitat [86]. Secondly, urbanization exerts intensive anthropogenic pressure on these areas. These sites are heavily modified and have a significant range of human events impacting them [87]. Soil erosion, trampling of vegetation, and soil compaction are three of the most visible forms of disruption [88]. These disruptions may promote the growth of alien and ruderal species, which can become common due to the additional nutrient supply and the high input of habitat loss and fragmentation propagules [89]. These findings were in accordance with the present results of tram and train tracks. The present study has shown that comparatively high anthropogenic disruption facilitates the coexistence of different plant forms encountered in urban flora, including two endemic and 60 alien species. This finding was also consistent with previous studies that the diversity of native plant species in less disturbed environments is relatively high, explaining the occurrence of several dominant species in the less disturbed habitats [90,91]. Additionally, native species diversity can quickly recover after being disturbed, but not if the disturbance is continuous [92]. The outcomes of the current study indicated that urban habitats under low urbanization intensity usually harbor a relatively higher richness of native species; therefore, they ought to be considered for the implementation of proper conservation measures.

## 5. Conclusions and Recommendations

This study highlighted the biotic components of the intra-city railway as an urban ecosystem in Alexandria. The range of habitats suitable for native species conservation was probably wider and more continuous in the less urbanized sites than in the urbanized ones. Thus, the loss of plant species could be induced by anthropogenic pressure. The train and tram vegetation could provide several beneficial functions. For instance, plants can positively affect the microclimate of habitats. An understanding of the floristic diversity on train and tram tracks is just as important for the possible future urban planning and land use management of cities. Native Mediterranean species predominate in intra-city railway flora; however, the contribution of alien and weed species is higher than their contribution in native floras. Spontaneous flora in intra-city railway areas exemplifies distinctive adaptations to unique urban ecosystems. The share of invasive species differs between intra-city railway and tram lines. Intra-city railway flora is species-rich; however, flora composition differs between urban habitats. The differences are related to disparities in abiotic and topographical factors, weather elements, and edaphic factors. Although the main proportion of the flora and species composition of the main transportation types, tram and train tracks, in the city included weeds, alien, and invasive species, the study revealed the occurrence of two endemic species (*Sinapis allionii* and *Sonchus macrocarpus*) with limited national distribution, as observed in other countries, especially in territories with high plant biodiversity [93,94]. This highlights the importance of these sites as refuges for some rare and endangered species worthy of conservation action. While managing urbanization impacts has become a conservation priority worldwide, the maintenance of Alexandria’s flora is often ignored. Egypt needs strategies for urban nature conservation that reflect the social demands on open spaces and environmental conditions.

## Figures and Tables

**Figure 1 biology-10-00698-f001:**
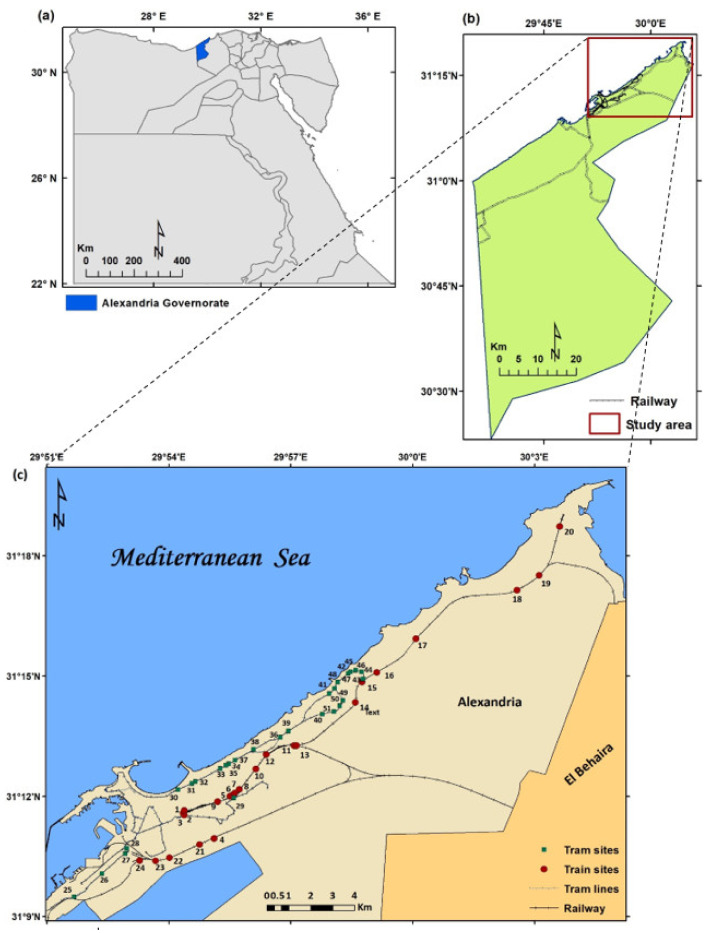
Location of the study area: (**a**) map of Egypt’s administrative boundaries; (**b**) location of Alexandria Governorate; and (**c**) study area, which includes the old City of Alexandria having the sampled sites along the tramlines and the railways in the city. All locations are georeferenced to geographical coordinates system (GCS).

**Figure 2 biology-10-00698-f002:**
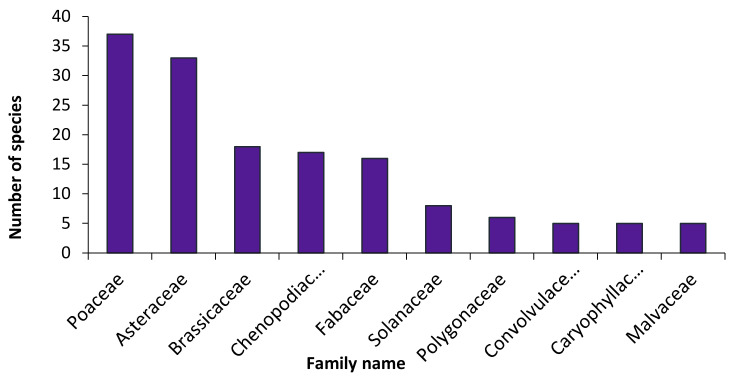
The families most represented in the present study.

**Figure 3 biology-10-00698-f003:**
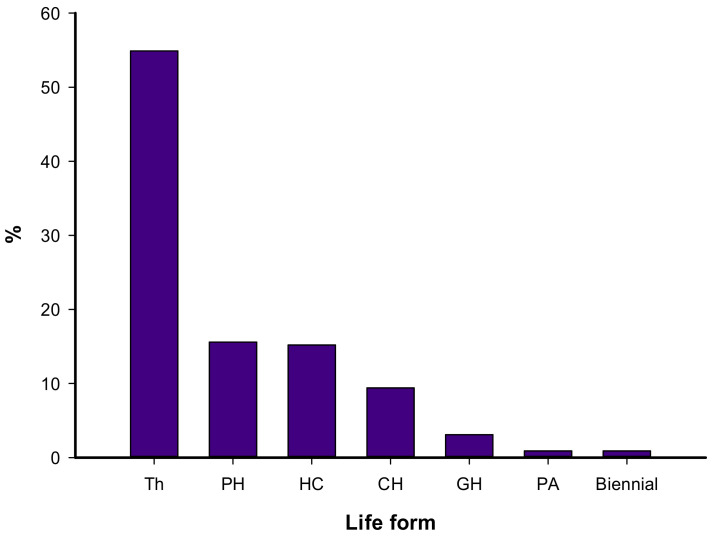
Lifeform spectrum of the recorded species in the study area. PH: Phanerophytes, CH: Chamaephytes, GH: Geophytes–Helophytes, HC: Hemicryptophytes, PA: Parasites, TH: Therophytes.

**Figure 4 biology-10-00698-f004:**
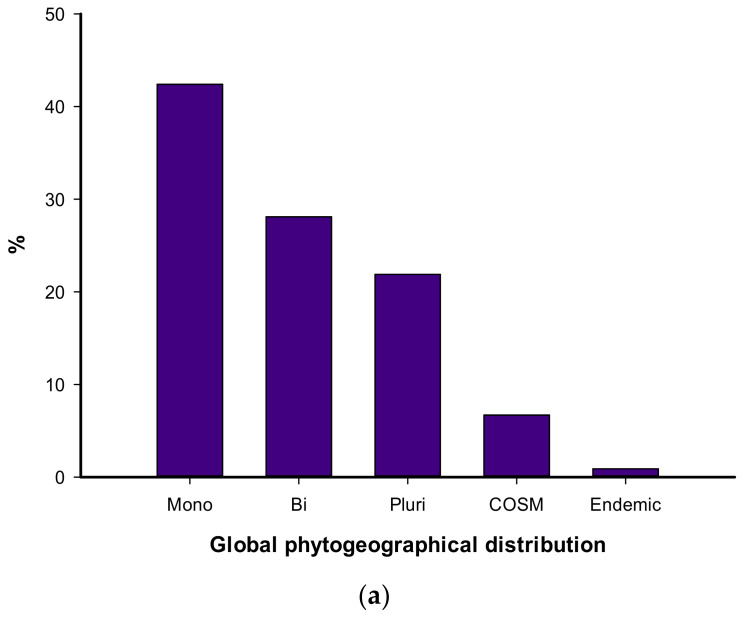
The global phytogeographical distribution of the species (**a**), chorotype spectrum of the recorded species in the study area (**b**). COSM: Cosmopolitan, Pluri: Pluri-regionals, SA: Saharo-Arabian, ME: Mediterranean, SU: Sudano-Zambezian, IT: Irano-Turanian, TR: Tropical, EU: Europian, PAN: Panotropic, PAL: Paleotropical.

**Figure 5 biology-10-00698-f005:**
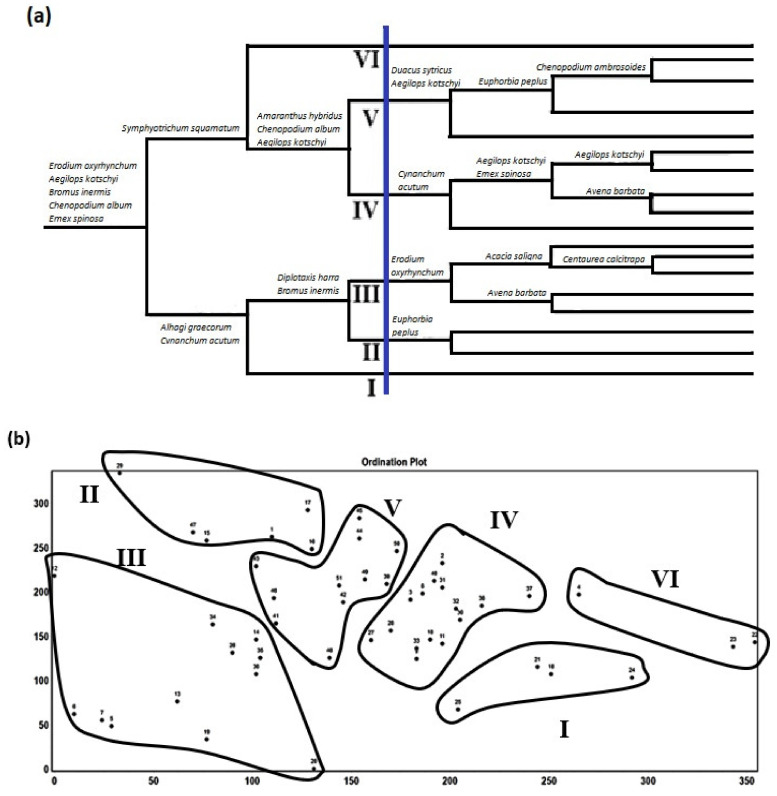
Dendrogram of the 6 vegetation groups derived after application of TWINSPAN classification technique (**a**), cluster segregation of the 6 vegetation groups along axes 1 and 2 using DECORANA (**b**). The 6 groups are named after characteristic species as follows: I: *Urospermum picroides*, II: *Chenopodium murale*, III: *Malva parviflora*, IV: *Cynodon dactylon*, V: *Hordeum leporinum*, and VI: *Sonchus oleraceus*.

**Figure 6 biology-10-00698-f006:**
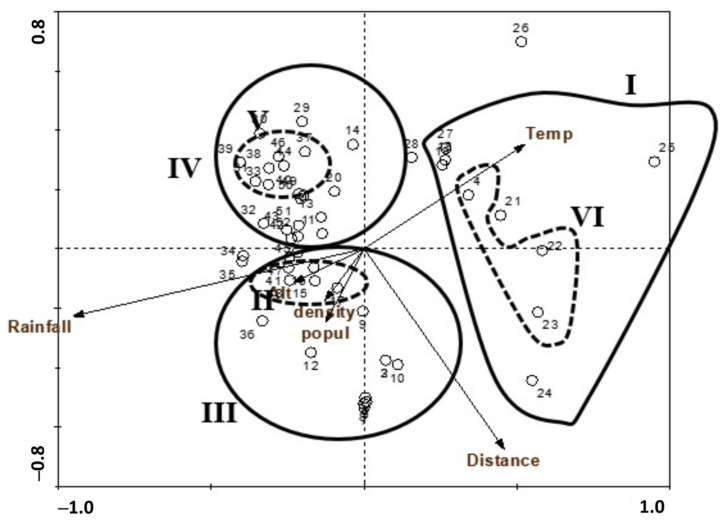
CCA biplot ordination of the sampled stands with environmental variables.

**Figure 7 biology-10-00698-f007:**
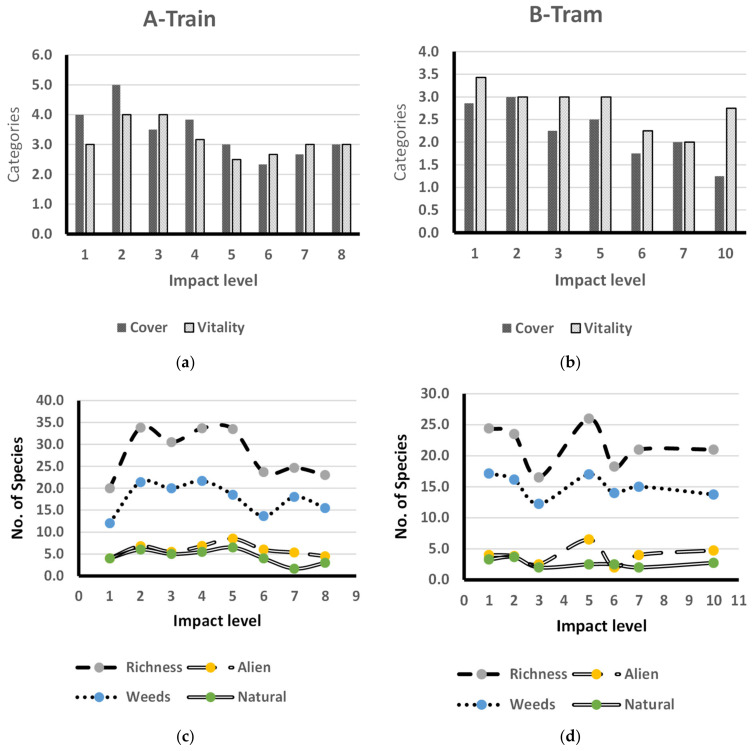
Species richness, number of alien species, weeds, natural species, cover, and species vitality across human impact levels along the train railway (**a**,**c**) and tram tracks (**b**,**d**).

**Table 1 biology-10-00698-t001:** Characteristics of the 6 vegetation groups derived after application of TWINSPAN. The vegetation groups are named as follows: I: *Urospermum picroides*, II: *Chenopodium murale*, III: *Malva parviflora*, IV: *Cynodon dactylon*, V: *Hordeum leporinum*, and VI: *Sonchus oleraceus*.

VG	No. of Stands	Train	Tram	First Dominant	Presence (%)	Second Dominant	Presence (%)
I	4	75	25	*Urospermum picroides*	83.3	*Sisymbrium irio*	83.3
II	6	66.7	33.3	*Chenopodium murale*	100	*Malva parviflora*	100
III	12	66.7	33.3	*Malva parviflora*	100	*Lactuca serriola*	100
IV	15	40	60	*Cynodon dactylon*	93.3	*Aegilops kotschyi*	93.3
V	11		100	*Hordeum leporinum*	100	*Phoenix dactylifera*	90.9
VI	3	100		*Sonchus oleraceus*	100	*Cynanchum acutum*	100

**Table 2 biology-10-00698-t002:** Components of plant diversity in the habitats and 6 vegetation groups identified in the present study. The vegetation groups are I: *Urospermum picroides*, II: *Chenopodium murale*, III: *Malva parviflora*, IV: *Cynodon dactylon*, V: *Hordeum leporinum*, and VI: *Sonchus oleraceus. F*-values represent one-way ANOVA, degrees of freedom (*df*) = 5. *t*-values represent Student’s *t*-test. **: *p* < 0.01, ***: *p* < 0.001, *ns*: not significant (i.e., *p* > 0.05).

Character		Richness	Simpson	Shannon	Sh. Even	Hill	Cover (%)	No. of Species	Natural Species	Native Weeds	Alien Species
Vegetation groups	I	20.50 ± 7.34	0.04 ± 0.02	2.77 ± 0.38	0.96 ± 0.01	1.80 ± 0.96	31.40 ± 25.49	58	6	33	19
	II	29.17 ± 11.70	0.03 ± 0.01	3.18 ± 0.34	0.96 ± 0.01	1.71 ± 0.78	73.58 ± 23.69	142	34	70	37
	III	29.25 ± 8.22	0.03 ± 0.01	3.21 ± 0.27	0.96 ± 0.01	1.43 ± 0.06	65.00 ± 7.07	72	14	42	16
	IV	25.80 ± 6.73	0.03 ± 0.01	3.09 ± 0.26	0.96 ± 0.01	1.63 ± 0.74	38.87 ± 22.20	104	24	58	22
	V	18.09 ± 2.98	0.04 ± 0.02	2.77 ± 0.18	0.88 ± 0.26	2.55 ± 3.05	16.55 ± 20.83	60	13	32	15
	VI	38.67 ± 15.82	0.03 ± 0.01	3.44 ± 0.39	0.95 ± 0.00	1.37 ± 0.03	61.67 ± 20.21	81	14	46	21
*F*-value		4.19 **	1.40 ^ns^	4.97 ***	0.67 ^ns^	0.62 ^ns^	8.77 ***	--	--	--	--
Habitat	Train	29.58 ± 10.91	0.03 ± 0.01	3.18 ± 0.35	0.96 ± 0.01	1.52 ± 0.49	59.77 ± 27.20	197	48	98	51
	Tram	21.56 ± 6.61	0.04 ± 0.02	2.91 ± 0.29	0.93 ± 0.17	2.12 ± 2.05	30.96 ± 26.26	139	26	72	41
*t*-value		10.36 ***	2.35 ^ns^	9.24 **	0.91 ^ns^	1.93 ^ns^	13.88 ***	--	--	--	--
Total mean		25.33 ± 9.69	0.03 ± 0.01	3.04 ± 0.34	0.94 ± 0.12	1.84 ± 1.55	44.17 ± 30.13	224	56	108	60

**Table 3 biology-10-00698-t003:** Mean ± standard deviation of the environmental characteristics in relation to habitats and 6 vegetation groups derived after application of TWINSPAN. The vegetation groups are I: *Urospermum picroides*, II: *Chenopodium murale*, III: *Malva parviflora*, IV: *Cynodon dactylon*, V: *Hordeum leporinum*, and VI: *Sonchus oleraceus*.

		Alt (m)	Population (km^−2^)	Rainfall (mm)	Temp. (°C)	Rd. Density (m/m^2^)	Distance (m)
VG	I	12.50 ± 14.38	29,736.0 ± 18,931.1	183.33 ± 0.82	20.33 ± 0.52	88.15 ± 42.05	8279.77 ± 7241.88
	II	47.83 ± 133.50	28,322.5 ± 12,730.0	182.67 ± 2.67	20.08 ± 0.29	99.55 ± 31.14	8478.83 ± 8137.16
	III	1.50 ± 1.91	25,715.5 ± 7216.3	177.00 ± 3.37	20.75 ± 0.50	83.42 ± 28.10	12,038.66 ± 11,759.29
	IV	17.33 ± 10.53	32,756.0 ± 14,661.1	182.67 ± 1.88	20.67 ± 0.49	125.17 ± 35.88	7565.04 ± 9019.00
	V	40.36 ± 28.88	51,816.9 ± 22,813.4	184.09 ± 0.30	20.00 ± 0.00	75.69 ± 17.41	630.76 ± 336.78
	VI	6.33 ± 6.81	30,949.0 ± 7175.9	178.67 ± 1.15	21.00 ± 0.00	65.79 ± 17.11	21,803.77 ± 1384.22
Habitat	Train	12.33 ± 13.03	27,263.3 ± 10,507.0	181.75 ± 2.13	20.38 ± 0.49	90.59 ± 36.27	15,363.05 ± 6638.39
	Tram	40.07 ± 89.25	41,518.6 ± 20,760.9	182.93 ± 3.06	20.37 ± 0.49	103.35 ± 35.24	618.20 ± 366.49
	Total	27.02 ± 66.45	34,810.2 ± 18,070.7	182.37 ± 2.71	20.37 ± 0.49	97.35 ± 35.95	7556.95 ± 8694.20

**Table 4 biology-10-00698-t004:** Simple linear correlation coefficient between diversity indices and environmental factors.

	Alt	Popul.	Cover	Rainfall	Temp	Rd. Density	Distance
Richness	0.112	–0.277 *	0.562 ***	–0.196	0.189	0.007	0.535 ***
Simpson	–0.225	0.335 *	–0.229	0.054	–0.247	–0.152	–0.354 **
Shannon	0.131	–0.328 *	0.589 ***	–0.211	0.205	0.009	0.498 ***
Sh. Even	–0.001	0.061	0.154	–0.075	0.111	0.083	0.123
Hill	0.089	–0.074	–0.290 *	0.164	–0.061	–0.019	–0.145

*: *p* < 0.05, **: *p* < 0.01, ***: *p* < 0.001.

## Data Availability

The data presented in this study are available on request from the corresponding author.

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
