# Peer review of "Pattern of Urban Flora in Intra-City Railway Habitats (Alexandria, Egypt): A Conservation Perspective"

_biology, 2021, doi:10.3390/biology10080698_

Round 1

Reviewer 1 Report

Please see attached pdf document.

Author Response

  1. July 2021

Ms. Cristina Maria Fiat

Assistant Editor

Biology

Dear Ms. Fiat,

Please find attached the revised manuscript titled ‘Pattern of urban flora in intra-city railway habitats: A conservation perspective’. Ms. Ref. No.: biology-1301942, authored by Selim Z. Heneidy, Marwa Waseem A. Halmy, Soliman M. Toto, Sania K. Hamouda, Amal Fakhry, Laila M. Bidak, Ebrahem M. Eid, and Yassin M. Al-Sodany.  

On behalf of my co-authors, I thank you very much for giving us the opportunity to revise our manuscript. We appreciate the positive and constructive comments and suggestions provided by the reviewers on our manuscript. We have carefully studied the reviewers’ comments and have made revisions that are yellow highlighted in the revised version of the manuscript. We have tried our best to revise our manuscript according to the reviewers’ comments. Please find attached the revised version of our manuscript, which we would like to submit for your kind consideration. Once again, we would like to express our great appreciation to you and the reviewers for the comments on our manuscript.

Please find below our detailed responses to each of the points raised.

---------------------------------------------------------------------------------------------------------------------

Reviewer comments:

114-5: Trains and trams are frequently contrasted; it would help readers if each of these categories of transit were explicitly described

Response: We greatly appreciate your critical observations as well as your constructive and helpful comments. We hope that we could address your questions/comments by the explanations and revisions made in the manuscript. We add a sentience to describe these categories of transit.

---------------------------------------------------------------------------------------------------------------------

257-60: For those unfamiliar with the Raunkiaer scheme, it would be helpful to spend a little bit of the methods describing Raunkiaer’s life forms and why they are an informative way to summarize a vegetation community. I would also recommend explaining a bit more clearly what is meant by “global” and “national” geographic distributions. There seem to be geographical classification schemes at work in the background here, but many readers (like myself) not versed in the floristics literature may not be familiar with these schemes and the categories they entail. This creates some confusion when the categories (e.g. Mediterranean coast vs. Nile region, mono-regional vs. bi-regional elements, etc.) are brought up in the results and discussion sections

Response: Raunkiaer’s life forms and why they are an informative way to summarize a vegetation community were described in the text. We change global distribution to chorotype and national distribution to local distribution. 

---------------------------------------------------------------------------------------------------------------------

283: What does “vitality” mean in this context? It becomes an important criterion in your results, so it should be clearly defined.

Response: The mean of vitality was given in the text.

---------------------------------------------------------------------------------------------------------------------

315: Is “weed” used as a synonym for wild plant? It’s a bit odd to see the native plants characterized as weeds and then juxtaposed to exotic plants that are not characterized as weeds. It all depends, of course, on one’s definition of “weed,” but the equivocity of the word is a good reason not to use it. “Adventitious” or “early successional” or “colonizing” or “disturbance-loving” might be terms that convey the intended meaning.

Response: Weeds does not mean a synonym for wild plant. It means any wild plant that grows in an unwanted place.

---------------------------------------------------------------------------------------------------------------------

Figure 2: The caption says “number and %,” but no percentages are shown. It should also be explained that the figure only includes the top 10 or the 51 families found

Response: The (%) is deleted. The only top 10 families included in the figure to exclude the crowded of 51 families.

---------------------------------------------------------------------------------------------------------------------

Figures 3 and 4: There are good reasons not to use pie charts for data visualization, and 3D graphs should be avoided in almost all circumstances. Tree plots (probably the best choice) or stacked barplots would convey the same information more clearly

Response: The histogram was used for figures 3 and 4.

---------------------------------------------------------------------------------------------------------------------

Figure 5: Text is illegible

Response: The text font was increased to be readable.

---------------------------------------------------------------------------------------------------------------------

Table 2: I’d like to see these data visualized with boxplots (or something like that) and analyzed statistically so that we can infer whether there is reason to believe that certain vegetation groups or habitats differ significantly from others

Response: The data is too much, and it is very difficult to show all variables in one figure and also, we have 7 figures and only 4 tables. ANOVA-1 was calculated and added to diversity indices. Please see Table 2 and results.

---------------------------------------------------------------------------------------------------------------------

348-350: Can species belong to more than one of these regions? I’m trying to understand why the percentages add up to more than 100

Response: Yes, the species can belong to more than one of these regions.

---------------------------------------------------------------------------------------------------------------------

Table 3: Again, I think these data should be visualized somehow. At the very least, a color ramp could be applies to each column of the table so that the relative magnitude of each row is easier to perceive. Better still, though, would be something like a violin plot that shows both the mean/median and the shape of the distribution of each variable. It’s probably true that many of these variables are not normally distributed, and summarizing data with a mean and standard deviation doesn’t really mean anything if data are not normally distributed

Response: The color ramp was used.

---------------------------------------------------------------------------------------------------------------------

Table 4: This table is fine, but I think color ramps would improve it

Response: The color ramp was used.

---------------------------------------------------------------------------------------------------------------------

490-2: It’s odd to summarize HI with its maxima rather than some representation of its central tendency and dispersion.

Response: The comment was changed and improved.

---------------------------------------------------------------------------------------------------------------------

Figure 7: Sites are replicated within the levels of train/tram x low/moderate/high HI, right? What I expected to see here, then, is something like a boxplot or violin plot that shows the distribution of each variable across the replicated sites. As it is, it seems to represent some aggregate value, like the cumulative richness across all sites within each train/tram x low/moderate/high

Response: The figures was changed and improved.

---------------------------------------------------------------------------------------------------------------------

I hope the explanation given above adequately addresses the reviewers’ comment. I would appreciate if the revised version of our manuscript would be considered for publication in Biology.

Sincerely,

Yassin M. Al-Sodany

[Kafrelsheikh University]

[Botany Department, Faculty of Science, Kafrelsheikh University, Kafr El-Sheikh 33516, Egypt]

[Email address: yalsodany@yahoo.com]

Reviewer 2 Report

The authors propose a manuscript titled “Pattern of urban flora in intra-city railway habitats: A conservation perspective

The article is original, well structured and written. In particular, this study takes into consideration the Intra-city railway areas that can have a role in enhancing the diversity and dynamic of urban flora. In particular the authors highlight the role the overlooked railway habitats can provide in conservation of biodiversity through surveying the floristic composition and diversity along intra-city railway and tram tracks in Alexandria City. Have also evaluated, into the plant communities detected, the adaptations to urban-industrial ecosystems with different levels of anthropogenic disturbance. Very interesting the data that a despite the high level of disturbance, native species dominated the investigated habitats, including rare and endemic species. After these few suggestions, the work may be published.

Introduction

Well done and referenced. Few observations and suggestions

  • The authors declare “Despite the numerous studies that addressed the ruderal vegetation and alien species of the Egyptian habitats, no study has accounted for the flora of Egypt’s major urban centers such as great Cairo and Alexandria”. Are really sure? sometimes there is unpublished gray literature. In any case, if the authors are certain of the lack data what is written is ok.
  • I suggested to choise a reference/s for this statements: “The construction and use of tracks, roads, canals, railways, and airports have involved many changes, some of them are direct and others are indirect. Direct influences include the destruction of the existing habitats and the provision of new ones that have special characteristics. These have provided more or less continuous stretches of open habitats extending for hundreds of miles and forming a nation-wide network, with opportunities for rapid colonization and spread”.
  1. Materials and Methods

Well done.

  • In the figure 1 I don’t see the latitude and longitude and the legend. Please if possible increase the size of the numbers and specify in the text the geographical system adopted (e.g. WGS84?)
  1. Results

The figure and tabes are clear. Following the suggestion in the suggested way

  • Lines 317-322. In literature the botanic nomenclature and scientific names of plants is cited with the author. Please only for the first time when cited the scientific name use the complete and correct way. Also verify the scientific name. e.g. Lactuca serriola not seriolag.
  • Acacia saligna (Labill.) H.L. Wendl.
  • Bassia indica (Wight) A.J. Scott
  • Dalbergia sissoo ex DC.
  • Ipomoea carnea….
  • Prosopis juliflora……
  • Chenopodium murale…..
  • Cynodon dactylon ….
  • Malva parviflora…..
  • Urospermum picroides…..
  • Sisymbrium irio….
  • Lactuca serriola….
  • Phoenix dactylifera….
  • Senecio desfontanei …
  • Sonchus oleraceus…
  • Avena fatua…
  • Hordeum leporinum…
  • Lines 489-490. Choise a reference for this statement “Results revealed that human activities have distinct effects on the flora of railway habitats”.

Conclusion

The conclusion deserve to add some references of some crucial concept, as I complete in the following way.

  • Lines 711-714. Although the main proportion of the flora and species composition of main transportation ways tram and train tracks in the city included weeds, alien and invasive species, the study revealed the occurrence of two endemic species (Sinapis allionii and Sonchus macrocarpus) with limited national distribution, as observed in other countries especially in territories with high plant biodiversity [Perrino et al. 2018, Wagensommer et al. 2021].

The references to be added:

  • Perrino, E.V.; Silletti, G.N.; Erben, M.; Wagensommer, R.P. Viola cassinensis lucana(Violaceae), a new subspecies from Lucanian Apennine, southern Italy. Phyton, 2018, 58(2), 109-115. Doi: https://doi.org/10.12905/0380.phyton58(2)-2018-0109
  • Wagensommer, R.P.; Venanzoni, R. Geranium lucarinii nov. and re-evaluation of G. kikianum (Geraniaceae). Phytotaxa, 2021, 489, 252–262. https://doi.org/10.11646/phytotaxa.489.3.2

Reference. Please use the guidelines of the journal and consider doi if avalilable.

Author Response

  1. July 2021

Ms. Cristina Maria Fiat

Assistant Editor

Biology

Dear Ms. Fiat,

Please find attached the revised manuscript titled ‘Pattern of urban flora in intra-city railway habitats: A conservation perspective’. Ms. Ref. No.: biology-1301942, authored by Selim Z. Heneidy, Marwa Waseem A. Halmy, Soliman M. Toto, Sania K. Hamouda, Amal Fakhry, Laila M. Bidak, Ebrahem M. Eid, and Yassin M. Al-Sodany.

On behalf of my co-authors, I thank you very much for giving us the opportunity to revise our manuscript. We appreciate the positive and constructive comments and suggestions provided by the reviewers on our manuscript. We have carefully studied the reviewers’ comments and have made revisions that are yellow highlighted in the revised version of the manuscript. We have tried our best to revise our manuscript according to the reviewers’ comments. Please find attached the revised version of our manuscript, which we would like to submit for your kind consideration. Once again, we would like to express our great appreciation to you and the reviewers for the comments on our manuscript.

Please find below our detailed responses to each of the points raised.

---------------------------------------------------------------------------------------------------------------------

Reviewer comments:

The authors declare “Despite the numerous studies that addressed the ruderal vegetation and alien species of the Egyptian habitats, no study has accounted for the flora of Egypt’s major urban centers such as great Cairo and Alexandria”. Are really sure? sometimes there is unpublished gray literature. In any case, if the authors are certain of the lack data what is written is ok.

Response: We greatly appreciate your critical observations as well as your constructive and helpful comments. We hope that we could address your questions/comments by the explanations and revisions made in the manuscript. Yes, we are certain of the lack data.  

---------------------------------------------------------------------------------------------------------------------

I suggested to choise a reference/s for this statements: “The construction and use of tracks, roads, canals, railways, and airports have involved many changes, some of them are direct and others are indirect. Direct influences include the destruction of the existing habitats and the provision of new ones that have special characteristics. These have provided more or less continuous stretches of open habitats extending for hundreds of miles and forming a nation-wide network, with opportunities for rapid colonization and spread

Response: The reference No. 14 was used as a reference for these statements.

---------------------------------------------------------------------------------------------------------------------

In the figure 1 I don’t see the latitude and longitude and the legend. Please if possible increase the size of the numbers and specify in the text the geographical system adopted (e.g. WGS84?)

Response: The Figure 1 was modified accordingly.

---------------------------------------------------------------------------------------------------------------------

Lines 317-322. In literature the botanic nomenclature and scientific names of plants is cited with the author. Please only for the first time when cited the scientific name use the complete and correct way. Also verify the scientific name. e.g. Lactuca serriola not seriolag.

Response: Thanks for these suggestions which already followed.

---------------------------------------------------------------------------------------------------------------------

Lines 489-490. Choise a reference for this statement “Results revealed that human activities have distinct effects on the flora of railway habitats”.

Response: This is our results and we discussed this in the discussion section.

---------------------------------------------------------------------------------------------------------------------

Lines 711-714. Although the main proportion of the flora and species composition of main transportation ways tram and train tracks in the city included weeds, alien and invasive species, the study revealed the occurrence of two endemic species (Sinapis allionii and Sonchus macrocarpus) with limited national distribution, as observed in other countries especially in territories with high plant biodiversity [Perrino et al. 2018, Wagensommer et al. 2021].

Response: This sentence was improved, and these two references were added.

---------------------------------------------------------------------------------------------------------------------

I hope the explanation given above adequately addresses the reviewers’ comment. I would appreciate if the revised version of our manuscript would be considered for publication in Biology.

Sincerely,

Yassin M. Al-Sodany

[Kafrelsheikh University]

[Botany Department, Faculty of Science, Kafrelsheikh University, Kafr El-Sheikh 33516, Egypt]

[Email address: yalsodany@yahoo.com]

Reviewer 3 Report

See file in attachment

Author Response

  1. July 2021

Ms. Cristina Maria Fiat

Assistant Editor

Biology

Dear Ms. Fiat,

Please find attached the revised manuscript titled ‘Pattern of urban flora in intra-city railway habitats: A conservation perspective’. Ms. Ref. No.: biology-1301942, authored by Selim Z. Heneidy, Marwa Waseem A. Halmy, Soliman M. Toto, Sania K. Hamouda, Amal Fakhry, Laila M. Bidak, Ebrahem M. Eid, and Yassin M. Al-Sodany.

On behalf of my co-authors, I thank you very much for giving us the opportunity to revise our manuscript. We appreciate the positive and constructive comments and suggestions provided by the reviewers on our manuscript. We have carefully studied the reviewers’ comments and have made revisions that are yellow highlighted in the revised version of the manuscript. We have tried our best to revise our manuscript according to the reviewers’ comments. Please find attached the revised version of our manuscript, which we would like to submit for your kind consideration. Once again, we would like to express our great appreciation to you and the reviewers for the comments on our manuscript.

Please find below our detailed responses to each of the points raised.

---------------------------------------------------------------------------------------------------------------------

Reviewer comments:

Lines 140-141: Add the authors to the names of taxa cited for the first time in the text

Response: We greatly appreciate your critical observations as well as your constructive and helpful comments. We hope that we could address your questions/comments by the explanations and revisions made in the manuscript. The authors were added the names of taxa cited for the first time in the text.

---------------------------------------------------------------------------------------------------------------------

Line 141: Correct the names of the species: “Sarcocornia fruticosa and Suaeda pruinosa” instead of “Sarcocorinia fruticose and Suaeda pruinose''

Response: Sorry for this mistake. The names were corrected.

---------------------------------------------------------------------------------------------------------------------

Lines 190-191: annual?

Response: Annual means yearly or per year.

---------------------------------------------------------------------------------------------------------------------

Tab. 1: Aegilops kotschyi (add 'o')

Response: It was added.

---------------------------------------------------------------------------------------------------------------------

Line 263: Is there no recent publication? More than 20 years is very much for Iucn red listing.

Response: This version is the last version published by IUCN.

---------------------------------------------------------------------------------------------------------------------

Caption to Fig. 3: GH instead of GE

Response: It was replaced.

---------------------------------------------------------------------------------------------------------------------

Line 358:' two endemic species' instead of 'two as endemic species'

Response: It was corrected.

---------------------------------------------------------------------------------------------------------------------

Caption to Fig. 4a: Add the meaning of 'pluri, cosm', etc

Response: It was added.

---------------------------------------------------------------------------------------------------------------------

Figure 5a: the names of the species are too small written and can not be read

Response: They were enlarged.

---------------------------------------------------------------------------------------------------------------------

Line 559: due to the evaluation

Response: It was added.

---------------------------------------------------------------------------------------------------------------------

Table S1: Why in red? (see Stand no. 25 and 26)

Response: The red was changed to black.

---------------------------------------------------------------------------------------------------------------------

Table S2: Add the caption to the Table

Response: The caption was added.

---------------------------------------------------------------------------------------------------------------------

I hope the explanation given above adequately addresses the reviewers’ comment. I would appreciate if the revised version of our manuscript would be considered for publication in Biology.

Sincerely,

Yassin M. Al-Sodany

[Kafrelsheikh University]

[Botany Department, Faculty of Science, Kafrelsheikh University, Kafr El-Sheikh 33516, Egypt]

[Email address: yalsodany@yahoo.com]
